# Comparative Effects of *Gymnema sylvestre* and Berberine on Adipokines, Body Composition, and Metabolic Parameters in Obese Patients: A Randomized Study

**DOI:** 10.3390/nu16142284

**Published:** 2024-07-16

**Authors:** Cindy Bandala, Jazmín Carro-Rodríguez, Noemí Cárdenas-Rodríguez, Itzel Peña-Montero, Modesto Gómez-López, Ana Paola Hernández-Roldán, Juan Carlos Huerta-Cruz, Felipe Muñoz-González, Iván Ignacio-Mejía, Brayan Domínguez, Eleazar Lara-Padilla

**Affiliations:** 1Laboratorio de Neurociencia Traslacional Aplicada a Enfermedades Crónicas y Emergentes, Escuela Superior de Medicina, Instituto Politécnico Nacional, Mexico City 11340, Mexico; crodriguezba@ipn.mx (C.B.); jazmincarrorodriguez@gmail.com (J.C.-R.); paolahdz0422@gmail.com (A.P.H.-R.); felipemgonzalez1@yahoo.com.mx (F.M.-G.); braydodope@gmail.com (B.D.); 2Laboratorio de Neurociencias, Instituto Nacional de Pediatría, Mexico City 04530, Mexico; noemicr2001@yahoo.com.mx; 3Laboratorio de Obesidad, Escuela Superior de Medicina, Instituto Politécnico Nacional, Mexico City 11340, Mexico; itzelpm.esm@hotmail.com (I.P.-M.); mgomezlo@ipn.mx (M.G.-L.); 4Unidad de Investigación en Farmacología, Instituto Nacional de Enfermedades Respiratorias, Ismael Cosio Villegas, Secretaria de Salud, Mexico City 14080, Mexico; juanc.huerta@iner.gob.mx; 5Laboratorio de Diseño y Desarrollo de Nuevos Fármacos e Innovación Biotecnológica, Mexico City 11340, Mexico; 6Laboratorio de Medicina Traslacional, Escuela Militar de Graduados en Sanidad, Universidad del Ejército y Fuerza Aérea, Mexico City 11200, Mexico; ivanignacio402@gmail.com

**Keywords:** *Gymnema sylvestre*, berberine, obesity, fasting glucose, body composition, adipokine, gene expression, resistin, omentin, visfatin, apelin

## Abstract

*Gymnema sylvestre* (GS) and berberine (BBR) are natural products that have demonstrated therapeutic potential for the management of obesity and its comorbidities, as effective and safe alternatives to synthetic drugs. Although their anti-obesogenic and antidiabetic properties have been widely studied, comparative research on their impact on the gene expression of adipokines, such as resistin (Res), omentin (Ome), visfatin (Vis) and apelin (Ap), has not been reported. Methodology: We performed a comparative study in 50 adult Mexican patients with obesity treated with GS or BBR for 3 months. The baseline and final biochemical parameters, body composition, blood pressure, gene expression of Res, Ome, Vis, and Ap, and safety parameters were evaluated. Results: BBR significantly decreased (*p* < 0.05) body weight, blood pressure and Vis and Ap gene expression and increased Ome, while GS decreased fasting glucose and Res gene expression (*p* < 0.05). A comparative analysis of the final measurements revealed a lower gene expression of Ap and Vis (*p* < 0.05) in patients treated with BBR than in those treated with GS. The most frequent adverse effects in both groups were gastrointestinal symptoms, which attenuated during the first month of treatment. Conclusion: In patients with obesity, BBR has a better effect on body composition, blood pressure, and the gene expression of adipokines related to metabolic risk, while GS has a better effect on fasting glucose and adipokines related to insulin resistance, with minimal side effects.

## 1. Introduction

Obesity is the abnormal or excessive accumulation of adipose tissue [1]. This disease negatively affects almost all physiological functions of the body, increasing the possibility of suffering comorbidities such as type 2 diabetes mellitus (T2DM), cardiovascular diseases, different types of cancer, and a variety of musculoskeletal disorders and mental health problems [2]. Adipocytes are metabolically active and have endocrine, paracrine, and autocrine functions. In patients with obesity, adipose tissue expands, increasing the size of adipocytes (hypertrophy), modifying their function, and increasing local inflammation [3].

Adipocyte hypertrophy leads to the synthesis of adipokines that have different physiological functions. Some adipokines that play important roles in obesity and metabolic diseases are visfatin (Vis), resistin (Res), apelin (Ap) and omentin (Ome) [4]. An increase in the serum concentrations of Vis, Res, and Ap, and a decrease in Ome are related to greater metabolic damage in patients with obesity and T2DM [5,6,7,8,9]. Res, also called “adipocyte-specific secretory factor”, has been positively correlated with insulin resistance in patients with obesity and has even been considered a predictor of T2DM [10,11]. This relationship is attributed to Res, which triggers mitochondrial dysfunction in tissues related to glucose metabolism [12]. Vis is a proinflammatory adipocytokine, also called pre-ß cell colony-enhancing factor 1 (pBEF-1). Plasmatic levels correlate positively with intra-abdominal fat mass and may promote insulin resistance, vascular dysfunction and the activation of the nod-like-receptor–protein-3 (NLRP3) inflammasome complex [6,13,14,15,16]. Ap increases in individuals with obesity and it has been associated with the expansion of white adipose tissue [7,8]. In women, a positive correlation has been described between body mass index (BMI) and the Homeostatic Model Assessment for Insulin Resistance [17]. Ome is downregulated in patients with obesity and related metabolic disorders. A negative correlation has been demonstrated between serum Ome concentration and BMI, as well as insulin resistance index, leptin, plasma glucose and insulin [18,19]. Therefore, reducing the expression of Res, Vis, and Ap and increasing Ome in patients with obesity have become therapeutic targets.

The treatment of obesity includes nutritional, psychological, and pharmacological management [20]. Natural products such as *Gymnema sylvestre* (GS) and berberine (BBR) have been used in the management of obesity, showing promising results for certain biochemical and anthropometric parameters [21,22,23,24].

GS, also known as “meshashringi”, is a plant native to central and southern India [25]. Leaf extract has been used as a laxative, diuretic, and cough suppressant; it also has antioxidant, antimicrobial, antihypercholesterolemic, and hepatoprotective properties [26,27]. Its adjuvant effect has been reported in the treatment of T2DM, revealing how its active compounds can influence glucose regulation and improve insulin sensitivity [28,29]. Furthermore, its administration can have a positive effect on the general metabolic profile, including improving parameters such as the lipid profile and blood pressure [30]. Although direct evidence linking GS to adipokine modulation is limited, its role in improving metabolic health suggests that it could impact these signaling molecules [31]. BBR is an alkaloid derived from plants native to China, such as *Rhizoma coptidis*, *Cortex pellodendri* and *Hydrastis canadensis*. These plants are used to treat infectious diarrhea, inflammation, T2DM, nonalcoholic fatty liver disease, dyslipidemia, cardiovascular diseases, and obesity [32], and have been shown to improve insulin sensitivity and stimulate glucose uptake through the activation of AMP-activated protein kinase (AMPK) [33] and the Toll-like receptor 4 (TLR4) signaling pathway, making them promising agents for treating metabolic syndrome and cardiovascular risk associated with obesity [23,34]. BBR inhibits complex I of the respiratory chain, contributing to its antioxidant capability by activating the translocation of nuclear factor erythroid 2–related factor 2 (Nrf2), which increases the gene expression of superoxide dismutase (SOD), glutathione peroxidase (GPX), and reduced glutathione (GSH) [35]. All of these mechanisms suggest that BBR could influence the gene and protein expression of adipokines, which are closely related to metabolic regulation [36].

The aim of this research was to evaluate the comparative efficacy and safety of GS and BBR in the treatment of obese Mexican patients for 3 months, as well as their effect on the gene expression of Res, Vis, Ap, and Ome.

## 2. Materials and Methods

### 2.1. Trial Oversight

A comparative, prospective and randomized study was performed in the Obesity Care Program at the Higher School of Medicine. This study was carried out in full accordance with good clinical practice guidelines and the Declaration of Helsinki. The study was registered with the Research and Ethics Committee of the Higher School of Medicine (ESM. CE-01/7-12-2015) and Clinical Trials NCT06426966. All patients signed informed consent forms, and the information was protected through confidentiality agreements. Randomization was carried out using a table of random numbers.

### 2.2. Patients

We included 50 Mexican patients of both genders who were over 18 years of age, had a body mass index (BMI) greater than 30 kg/m^2^, and did not have a previous diagnosis of T2DM but presented at least two risk factors for the disease (history of parents or siblings with T2DM, over 40 years of age, sedentary lifestyle habits, controlled arterial hypertension, fasting blood glucose < 126 mg/dL, or glycated hemoglobin < 6.5%). The key exclusion criteria were (i) pregnant patients, (ii) depressed patients, (iii) patients with allergic reactions to any component of the supplements, and (iv) patients treated with any natural or synthetic anti-obesogenic or antidiabetic drug. Group A (25 patients) was treated with GS at a dose of two 200 mg capsules before breakfast, and Group B (25 patients) was treated with BBR at a dose of one 500 mg tablet three times a day before each meal. In both groups, the treatment lasted 3 months.

### 2.3. Trial Procedures and Outcomes

Anthropometric, biochemical and blood pressure parameters were measured before and after treatment. The anthropometric measurements included body weight, height, body mass index (BMI) according to the obesity classification of the World Health Organization [37], hip circumference, and waist–hip ratio (WHR), in addition to body composition, which was measured using the InBody 770. Blood pressure was measured using Welch Allyn brand sphygmomanometers, with cuffs for obese patients. Regarding biochemical parameters, fasting glucose, lipid profile (total cholesterol, triglycerides, HDL, LDL), basal insulin, and HbA1c were measured. The adherence to treatment and the presence of adverse effects were recorded using a log that noted the time at which the tablets were ingested and any adverse effects experienced that day.

Whole blood samples were taken, and total RNA (tRNA) was extracted using a TRIzol reagent, which consists of a mixture of guanidine isothiocyanate and phenol–chloroform. Once the total RNA was isolated, it was suspended in RNase-free water to prevent the possible degradation of the sample before reverse transcription. The extraction and integrity of the RNA were verified using agarose gel electrophoresis. The final purity of the samples was calculated based on the absorbance ratio at 260/280 nm. cDNA amplification was performed using the ‘First Strand cDNA Synthesis’ kit for RT–PCR from Roche. A real-time polymerase chain reaction (RT–PCR) procedure was performed to determine the relative expression of the mRNAs of the genes studied using probes from the Human Universal Probe Library, a LightCycler thermocycler, and a TaqMan reaction mixture, all from Roche Diagnostics (Roche Diagnostics GmbH, Mannheim, Germany). The oligo sequences of the primers (sense and antisense) were designed with ProbeFinder software (Ap, NM_017413.4, F, 5′ gaa agt ggg gga tgg cta ag 3′, R, 5′ ccc acc cac tac cct ctt ct 3′, Ome, NM_017625.2, F, 5′ tga ggg tca ccg gat gta ac 3′, R, 5′ gga ctg gcc tct gga aag ta 3′, Res, NM_001193374.1, F, 5′ cca ccg aga ggg atg aaa g 3′, R, 5′ ttc ttc cat gga gca cag g 3′ and Vis, NM_005746.2, F, 5′ aag gga tgg aac tac att ctt gag 3′, R, 5′ ctg tgt ttt cca ccg tga ag 3′). The reaction mixture was prepared according to the manufacturer’s instructions. Each sample was analyzed in duplicate, and the data obtained were analyzed using LightCycler Nano software, version 1.1.

### 2.4. Statistical Analysis

The distribution of the quantitative data was assessed by the Shapiro–Wilk test. The comparison of frequencies was carried out using the chi-square (X^2^) test, while the comparison of basal means between groups was performed with Student’s *t*-test. The final means were compared using Student’s *t*-test when no statistically significant differences were found in the basal comparison. For parameters with significant differences in the basal measurement, covariate adjustment (repeated-measures ANOVA) and the Bonferroni correction were applied to compare the final means between groups. Self-controlled analysis was performed with a paired samples *t*-test. Analyses were performed with GraphPad Prism software, version 8.0.0 for Windows (GraphPad Software, San Diego, CA, USA), and SPSS software, version 19 (IBM Corp. Released 2015. IBM SPSS Statistics for Windows, Version 19.0. Armonk, NY, USA: IBM Corp.). A value of *p* < 0.05 was considered to indicate statistical significance.

## 3. Results

### 3.1. Clinical and Demographic Data

This study included 50 patients diagnosed with exogenous obesity. As shown in Table 1, both treatment groups were homogeneous (*p* > 0.05) in terms of age, gender, and type of obesity.

### 3.2. Follow-Up and Outcomes

#### 3.2.1. Principal Outcomes

Figure 1 shows the gene expression of Res (1A-B), Ome (1C-D), Vis (1E-F) and Ap (1G-H) before and after treatment with GS and BBR. The self-controlled analysis demonstrated that GS significantly decreased (*p* < 0.05) the gene expression of Res (Figure 1A), while BBR significantly increased the expression of Ome (Figure 1D) and decreased the gene expression of Vis (Figure 1F) and Ap (Figure 1H). A comparative analysis of the final values of BBR vs. GS revealed that BBR significantly decreased (*p* < 0.05) the gene expression of Vis (Appendix A) and Ap (Appendix A).

#### 3.2.2. Secondary Outcomes

According to our self-controlled analysis, GS significantly decreased (*p* < 0.05) the body fat percentage, fasting glucose, and glycosylated hemoglobin, and increased basal insulin (Appendix A). BBR significantly decreased (*p* < 0.05) body weight, BMI, body fat percentage, visceral fat percentage, and diastolic blood pressure (Appendix A).

Table 2 shows the significant differences (*p* < 0.05) in BMI, WHR, systolic blood pressure (SBP), and diastolic blood pressure (DBP) between the treated patients in the BBR group and those in the GS group (Table 2).

As shown in Table 3, a significant difference (*p* < 0.05) was observed in the decrease in fasting glucose in the group of patients treated with GS compared to the group treated with BBR.

#### 3.2.3. Adverse Effects of *Gymnema sylvestre* and Berberine

Patients who were treated with GS reported more gastrointestinal adverse effects (dysgeusia, 40%; diarrhea, 28%; nausea, 20%; reflux, 28%; decreased consistency of stools, 52%; and polydipsia, 36%) than patients treated with BBR (nausea, 20%; constipation, 16%; and hemorrhoidal bleeding, 28%) (Appendix A). Patients who were treated with GS reported more adverse somatic (general malaise, sleep disturbances, and insomnia), vascular (vasculopathy and palpitations), and sexual (increased libido and breast turgor) effects (Appendix A). In both groups, these symptoms decreased within the first month of treatment.

## 4. Discussion

Overweight and obesity are diseases with increasing prevalence worldwide and are among the most significant public health problems [1]. The functions of adipose tissue extend beyond lipid storage to include immunological and endocrine roles. The endocrine function of adipose tissue is mainly regulated by the secretion of adipokines such as adiponectin, leptin, Ap, Ome, Res, and Vis, which are involved in several metabolic regulatory processes. An increase or decrease in these adipokines can cause metabolic irregularities [38]. Ome is released from adipocytes in visceral fat, and plasma levels are lower in individuals with cardiovascular risk, obesity, overweight, and T2DM [39,40,41]. Ap functions as an endogenous ligand for the G protein-coupled Ap receptor (APJ) and is mainly localized in adipose tissue. It plays a central role in processes such as apoptosis and inflammation, and its levels are higher in individuals with obesity and T2DM [42,43,44,45]. Res is a signaling polypeptide that is overexpressed in obese and insulin-resistant mice, promotes inflammation, and its expression is reduced with sensitizing drug treatment and weight loss, suggesting that Res links obesity and T2DM [46,47,48]. Vis, an insulin-mimetic adipokine, is produced in visceral adipose tissue and other sources and is considered a proinflammatory cytokine, elevated in obese and insulin-resistant individuals [49,50,51,52,53,54].

In the primary outcomes of our study, we analyzed the effects of GS and BBR on the gene expression of Res, Vis, Ap, and Ome in subjects with obesity. Our results showed that GS reduced the gene expression of Res, while BBR lowered the gene expression of Vis and Ap and enhanced Ome (Figure 1). These results can be explained by GS’s effect on glucose metabolism parameters, while BBR enhances lipid metabolism and fatty acid oxidation, and reduces adipogenesis and insulin resistance in adipocytes, thereby alleviating obesity and inflammation [55,56,57,58]. Both effects lead to a reduction in the proinflammatory and pro-oxidant state of adipose tissue, lowering the production of harmful adipokines (Res, Vis, and Ap) [59,60,61,62,63], and an elevation in protective adipokines, which is associated with the inhibition of inflammatory mediators related to the NF-κB pathway and an increase in the nuclear translocation of Nrf2 (redox control), endothelial protection, and glucose homeostasis in individuals with obesity and T2DM [18,64,65]. In this sense, recent studies have shown a negative correlation with serum Ome levels and a positive correlation with serum Res, Vis and Ap levels and with body weight, BMI, fat-free mass, insulin, and glucose levels in obese subjects [11,13,17,19,66]. The ability of both GS and BBR to lower the gene expression of inflammation-related adipokines (Ap, Res, and Vis) may explain the anti-obesity effects of these natural products [67].

For the secondary outcomes, our study showed that patients treated with GS experienced significant reductions in body fat percentage, glycosylated hemoglobin, and fasting glucose concentration, along with a rise in insulin, compared to baseline levels (Appendix A). These findings align with the current literature, showing that GS’s hypoglycemic activity primarily enhances insulin secretion in the pancreas [68], which lowers plasma glucose and glycosylated hemoglobin levels and raises serum insulin concentrations [69,70]. Other proposed mechanisms include the inhibition of intestinal glucose absorption and the increased sensitivity of peripheral insulin receptors, leading to enhanced intracellular glucose transport [71,72]. Higher insulin concentrations may compensate for lower blood glucose levels, while enhanced insulin sensitivity improves tissue glucose uptake, reducing the need for elevated insulin levels [73]. An in vivo study using oral administration of a GS extract, Om Santal Adivasi (OSA^®^) (1 g/d for 60 days), showed significant elevations in circulating insulin and C-peptide, which were associated with declines in fasting and postprandial blood glucose levels [74]. Another study showed that the administration of GS (300 mg b.i.d.) reduced glucose tolerance and glycosylated hemoglobin levels while increasing insulin sensitivity [75]. Our study also showed that GS significantly reduced patient weight. Several in vitro studies with different doses and administration times of GS (20–200 mg/kg for up to 30 days) have shown a decrease in BMI and visceral fat [26,76,77]. This effect may be explained by the ability of GS to reduce sweet cravings and control glucose and triglycerides [78]. In human studies, GS capsules (1 g) taken twice daily for 30 days were shown to reduce glucose, triglycerides, cholesterol, and low-density lipoprotein (LDL) levels in patients with T2DM [79]. The use of a water-soluble extract of GS leaves (400 mg/day) in patients with T2DM resulted in lower levels of glycosylated hemoglobin and glycosylated plasma protein than in controls, suggesting that GS increases endogenous insulin, possibly by regenerating the remaining β-cells in T2DM [80]. Another study showed that GS (400 mg for 8 weeks) combined with bioavailable (-)-hydroxycitric acid and niacin-bound chromium (4 mg for 8 weeks) lowered body weight, BMI, food intake, total cholesterol, LDL, triglycerides, and serum leptin levels [81]. GS has also been proposed as a nutraceutical therapy for metabolic syndrome [82]. A study in patients with glucose intolerance showed that GS at a dose of 300 mg twice daily (b.i.d.) significantly reduced body weight, BMI, and LDL [75]. In a study of patients with BMIs between 30 and 40 and fasting glucose levels between 100 and 125 mg/dL, daily supplementation with two sachets, each containing 1950 mg of myo-inositol, 50 mg of d-chiro-inositol, 50 mg of α-lactalbumin, and 250 mg of GS for 6 months, improved the fasting insulin level, homeostasis index, blood glucose level, lipid profile, BMI, weight loss, and waist circumference. These effects are likely due to GS’s role in regulating lipid and carbohydrate metabolism and altering the taste receptors on the tongue, thereby reducing sweet taste perception and intensity [24,28]. GS also exhibited anti-inflammatory and antioxidant activities via the modulation of the NF-κB/MAPK pathway through its active derivatives, including gymnemic acid, gymnema saponins, gymnemoside, gymnemasin, quercetin, and long fatty acids. A recent study showed that GS supplementation modifies the expression of genes related to insulin, lipid and carbohydrate metabolism, oxidative stress, and inflammation in the pancreas and liver of alloxan-induced hyperglycemic rats [70]. These findings may explain its effect on inflammation in obese subjects and help establish metabolic control [62]. These results, together with the decrease in Res gene expression, mainly demonstrate the antihyperglycemic properties of GS in obese patients.

Our study found that BBR significantly reduced body weight, BMI, body fat percentage, visceral fat percentage, and diastolic blood pressure compared to baseline, though these reductions were not significant for glucose regulation parameters (Appendix A). These results are consistent with clinical trials showing that BBR effectively lowers lipid levels (plasma triglycerides, total cholesterol, and LDL) [83,84,85] but has a limited effect on glucose metabolism. BBR does not have an insulin secretagogue effect on β-TC3 cells in vitro but may have a glucose-lowering effect on hepatocytes (HepG2 cell line) [86]. However, another study in poorly controlled diabetic patients receiving insulin injections showed that BBR increased fasting and postprandial C-peptide levels (a marker of pancreatic β-cell function) [83]. This suggests that BBR may not directly stimulate insulin secretion but may improve islet function in patients who do not respond to oral hypoglycemic agents [87]. However, its hypoglycemic effect is still controversial, since other studies have shown that the administration of BBR (0.5 g t.i.d.) at the beginning of each meal reduces fasting and postprandial blood glucose in patients with newly diagnosed T2DM [83]. Another study reported that glycosylated hemoglobin decreased after BBR treatment (0.5 g b.i.d.), and there were significant improvements in blood glucose and lipid levels [88]. In obese and diabetic patients, BBR combined with silymarin reduced fasting blood glucose and insulin, the insulin resistance index, HDL and LDL, triglycerides, uric acid, BMI, and the WHR [89]. The effect of BBR on the decrease in biochemical and anthropometric parameters could be explained by its anti-inflammatory and lipid-modifying effects in patients. Recent research has shown that BBR (at 50 and 100 mg/kg) in a tilapia model reduced plasma lipid levels but increased the expression of the peroxisome proliferator-activated receptor α (*ppar-α*) and carnitine palmitoyltransferase 1 (*cpt-1*) genes in the liver, leading to reduced lipid accumulation. In addition, BBR increased the gene expression of the antioxidant components erythroid 2-related factor 2 (*nrf-2*), heme oxygenase 1 (*ho-1*), and glutathione-S-transferase-α (*gst-α*), and reduced inflammation by decreasing Toll-like receptor-2 (*tlr-2*), myeloid differentiation protein-88 (*myd-88*), interleukin-1β (*il-1β*), tumor necrosis factor-α (*tnf-α*), and interleukin-8 (*il-8*) gene expression. The protective effect of BBR on metabolic function, which regulates lipid metabolism, antioxidant status, and the immune response, may be attributed to the modulation of the Nrf2, NF-κB, and PPAR-α pathways [90]. BBR also reduced systolic and diastolic blood pressure in our patients. These findings can be explained by BBR acting as a vasodilator in isolated blood vessels by reducing nicotinamide adenine dinucleotide phosphate oxidase (NOX)-2 and -4, extracellular signal-regulated kinase 1/2 (Erk1/2), inducible nitric oxide synthase (iNOS), and induced Cu/Zn-superoxide dismutase (Cu/Zn-SOD) levels [91].

Finally, adverse effects during drug administration were reported in our study groups (Appendix A). More symptoms were observed in the GS-treated group, whereas adverse effects were rare in the BBR-treated group. Gastrointestinal symptoms, mainly diarrhea and nausea, were the most common symptoms in both groups. The possible adverse effects may be caused by an increase in the glucose concentration in the intestinal lumen, leading to an osmotic effect and consequently to diarrhea [92]. In addition, a severe decrease in glucose may cause hypoglycemic effects such as nausea, which may lead to vomiting. Conversely, GS has adverse effects such as hypersensitivity, significant hypoglycemic effects, and drug interactions [93]. Another study reported that only mild gastrointestinal adverse events, such as constipation, flatulence, and diarrhea, were observed with BBR administration [88]. These findings are consistent with those of the present study. Our results suggest that GS and BBR could benefit patients with obesity and T2DM when prescribed individually or in combination; however, studies are required to evaluate their potential synergy in the management of patients with obesity.

## 5. Conclusions

In our study, GS and BBR showed therapeutic potential for the treatment of patients with obesity, with minimal adverse effects that decreased as the patient adapted. After three months of treatment, body fat percentage and fasting glucose decreased, and insulin levels increased in the group of patients treated with GS compared to the basal levels, while patients treated with BBR had a decrease in body weight, BMI, and body and visceral fat percentage compared to baseline. The novelty of our study is that we show that BBR decreases the gene expression of Vis and Ap and increases that of Ome, while GS decreases the gene expression of only Res in patients with obesity, coinciding with their respective effects on biochemical and anthropometric parameters.

We conclude that BBR is more effective than GS in the treatment of patients with obesity in terms of body composition parameters, blood pressure, and related adipokines (Vis, Ap, and Ome), while GS may have a better effect on reducing fasting glucose and its related adipokine Res. Based on these results, we propose that GS and BBR can complement the management of patients with obesity and T2DM with minimal side effects.

However, future studies are needed to determine liver and kidney function, the effect of other important adipokines, and the possibility of synergistic effects when these factors are administered simultaneously, along with a long-term follow-up of obese patients. 

## Figures and Tables

**Figure 1 nutrients-16-02284-f001:**
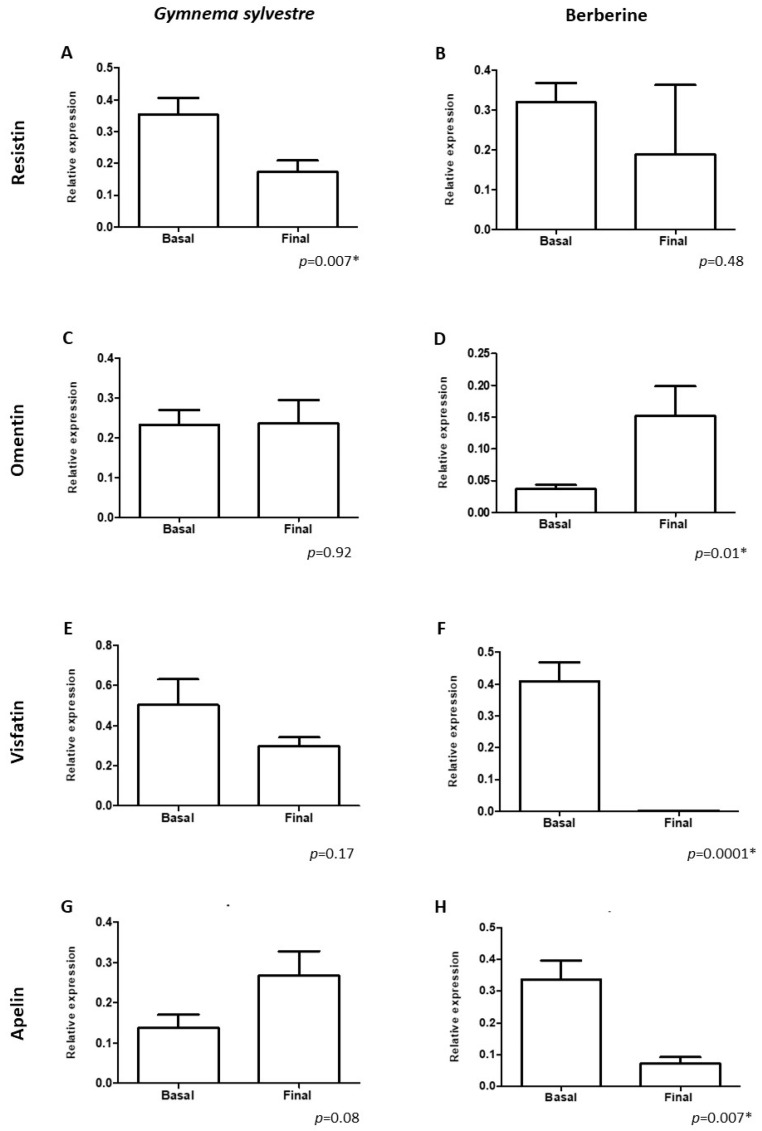
Gene expression of resistin (**A**,**B**), omentin (**C**,**D**), visfatin (**E**,**F**), and apelin (**G**,**H**) before and after treatment with *Gymnema sylvestre* and berberine. * Statistical significance.

**Table 1 nutrients-16-02284-t001:** Comparison of gender, age, and obesity type in relation to the treatment group.

	*Gymnema silvestre* (*n* = 25)	Berberine (*n* = 25)	*p* Value
Gender			
Women	84% (21)	84% (21)	0.64
Age (Mean ± SE)	39 ± 2.22	41 ± 2.24	0.58
Obesity classification ^&^			
Class I	52% (13)	60% (15)	0.30
Class II	32% (8)	40% (10)	
Class III	16% (4)	-	

SE = Standard error, ^&^ World Health Organization.

**Table 2 nutrients-16-02284-t002:** Mean comparison of the anthropometric and physiological measurements according to the treatment group.

	*Gymnema silvestre* Mean ± SE	Berberine Mean ± SE	*p* Value
Body Weight (kg)	93.18 ± 19.31	86 ± 11.06	0.11
BMI (kg/m^2^)	36.18 ± 6.22	31.53 ± 7.12	0.002 *^a^
WHR (cm)	0.98 ± 0.08	0.90 ± 0.06	0.001 *
Body fat (BF%)	46.93 ± 4.56	44.07 ± 4.80	0.54 ^b^
Visceral fat (VF%)	20.56 ± 3.91	44.07 ± 4.80	0.72 ^c^
SBP (mmHg)	119.20 ± 12.22	112.72 ± 7.67	0.02 *
DBP (mmHg)	83.20 ± 11.53	72.80 ± 7.91	0.003 *^d^

SE = Standard error, BMI = body mass index, WHR = waist-hip ratio, SBP = systolic blood pressure, DBP = diastolic blood pressure. The covariates that appear in the model were evaluated with the following value: ^a^. BMI = 34.5, ^b^. BF% = 46.7, ^c^. VF% = 33.14, ^d^. DBP = 80. * Statistical significance.

**Table 3 nutrients-16-02284-t003:** Mean comparison of the biochemical measurements in relation to the treatment group.

	*Gymnema silvestre* Mean ± SE	Berberine Mean ± SE	*p* Value
Fasting glucose (mg/dL)	86.43 ± 7.85	116.44 ± 7.37	0.0001 *^a^
Insulin (μU/mL)	19.43 ± 13.80	15.76 ± 13.50	0.34
Glycosylated hemoglobin (%)	5.50 ± 0.28	5.68 ± 0.25	0.40
Cholesterol (mg/dL)	190.97 ± 39.09	185.46 ± 47.23	0.65
Triglycerides (mg/dL)	127.99 ± 57.17	141 ± 45.33	0.37
LDL (mg/dL)	101.64 ± 25.19	98.96 ± 34.39	0.75
HDL (mg/dL)	40.95 ± 11.10	35.10 ± 7.34	0.57 ^b^
VLDL (mg/dL)	48.37 ± 20.69	51.39 ± 26.65	0.65

SE = Standard error, LDL = low-density lipoprotein, HDL = high-density lipoprotein, VLDL = very low-density lipoprotein. The covariates that appear in the model were evaluated with the following value: ^a^. Fasting glucose = 101.94, ^b^. HDL = 38. * Statistical significance

## Data Availability

The data presented in this study are available upon request to the corresponding author since they are clinical data of patients protected under a confidentiality letter delivered to the institutional ethics committee.

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
