# Peer review of "Comparative Effects of Gymnema sylvestre and Berberine on Adipokines, Body Composition, and Metabolic Parameters in Obese Patients: A Randomized Study"

_nutrients, 2024, doi:10.3390/nu16142284_

Round 1
Reviewer 1 Report
Comments and Suggestions for Authors
Manuscript: “Comparison of the efficacy of Gymnema sylvestre and Berberine on the gene expression of adipokines and clinical outcomes in patients with exogenous obesity”
The manuscript submitted by bandala et al. provides an evaluation of the effects of Gymnema sylvestre and Berberine in the expression of adipokines in a population of 50 adult Mexican patients with obesity. It also relates levels of expression of those adipokines with clinical outcomes of those patients.
Major comments:
i) The abstract needs considerable improvement to provide a more precise representation of the work presented in the manuscript.
ii) The rational for the choice of the adipokines is not appropriately described in the manuscript, and the itemization used in introduction for referring to each of the 4 adipokines (Res, Vis, Ap, Ome) is most definitely not the most appealing for the reader. The authors should integrate in a better mode each of the adipokines in the introduction section.
iii) The whole Materials and Methods section would benefit from
iv) Why was blood pressure the only physiological parameter evaluated?
v) The authors do not provide the rational for the chosen dosages of GS and BBR nor for the duration of three months used in the study.
vi) Cannot understand the need for the presentation of the comparative figure 3. In fact, a unique figure with the four panels showing the effects of Gs and BBR in the levels of the 4 adipokines would suffice. Having several panels repeating information with different scales does not help.
vii) The discussion section is very confusing and requires considerable language editing but as well careful association of points raised by authors and literature. As an example, in discussion section the lowering of plasma glucose caused by GS supplementation is discussed using various literature studies, however the presentation of such discussion should be more carefully made. First, references are not presented immediately for some of the points raised (e.g., reparative effect in pancreatic cells); second, there is a simultaneous reference to increased insulin concentrations and increased sensitivity of peripheral insulin receptors, effects that are somewhat controversial; why the need for increased insulin together with increase in insulin sensitivity. The way it is written makes it very hard to follow the line of reasoning. Series of studies are mentioning but without establishment of a line of questioning and answering.
viii) The effects of GS and BBR in adiponectin’s are discussed mostly taking into consideration studies that measured pro-inflammatory/anti-inflammatory cytokines or other effectors. Why did the authors not measure such factors in plasma from the patients to have a direct inference instead of just referring to literature results?
i) Page 3 line 31 – “In the first study group”; there is no need to start the sentence with this expression; the same for the following expression “while in the second group”. Needs language editing the sentence.
ii) Page 4 lines 150, 151: “Why mentioning “Additionally”; were samples of blood collected differently for analysis performed? Do not see the need for the inclusion of such sentence in the manuscript.
Comments on the Quality of English LanguageThroughout the text there are considerable issues with sentences organization. That deters the reader from an intelligible understanding of the manuscript. In particular the discussion section and some narrative parts of the introduction.
Author Response
The manuscript submitted by Bandala et al. provides an evaluation of the effects of Gymnema sylvestre and Berberine in the expression of adipokines in a population of 50 adult Mexican patients with obesity. It also relates levels of expression of those adipokines with clinical outcomes of those patients.
Major comments:
1. The abstract needs considerable improvement to provide a more precise representation of the work presented in the manuscript.
RESPONSE: We appreciate your comment, and we have improved the abstract as you suggested.
2. The rational for the choice of the adipokines is not appropriately described in the manuscript, and the itemization used in introduction for referring to each of the 4 adipokines (Res, Vis, Ap, Ome) is most definitely not the most appealing for the reader. The authors should integrate in a better mode each of the adipokines in the introduction section.
RESPONSE: We have improved the writing of the introduction as you suggested.
3. The whole Materials and Methods section would benefit from:
3.1 Why was blood pressure the only physiological parameter evaluated?
RESPONSE: Hypertension is a common comorbidity in patients with obesity, so we were interested in evaluating the effect of GS and BBR on these parameters in our population. The effect of BBR on hypertension is contradictory (DOI: 10.1016/j.ctcp.2020.101287, DOI: https://doi.org/10.1530/JME-17-0014), while in the case of GS, has been found to affect diastolic blood pressure (https://doi.org/10.1002/ptr.7585). In the Materials and Methods section, we replaced “physiological measurements” with “blood pressure measurements”.
3.2 The authors do not provide the rational for the chosen dosages of GS and BBR nor for the duration of three months used in the study.
RESPONSE: The doses and timing selected were based on clinical studies that reported adequate efficacy and adverse effects. For example, Joffe and Freed administered GS to patients with diabetes mellitus at a dosage of 400 mg per day for a period of 3 months. Jun Yin et al. administered 500 mg BBR three times a day for 3 months to adults with type 2 diabetes.
Joffe, D.J.; Freed, S.H. Effect of extended release Gymnema sylvestre leaf extract alone or in combination with oral hypoglycemic or insulin regimens for type 1 and type 2 diabetes. Diabetes Control Newsletter. 2001; 76: 30.
Jun Yin, Huili Xing, Jianping Ye. Efficacy of Berberine in Patients with Type 2 Diabetes. Metabolism. 2008;57(5): 712–717.
4. Cannot understand the need for the presentation of the comparative figure 3. In fact, a unique figure with the four panels showing the effects of Gs and BBR on the levels of the 4 adipokines would suffice. Having several panels repeating information with different scales does not help.
RESPONSE: We appreciate your comment and have made the changes you suggested.
5. The discussion section is very confusing and requires considerable language editing but as well careful association of points raised by authors and literature. As an example, in discussion section the lowering of plasma glucose caused by GS supplementation is discussed using various literature studies, however the presentation of such discussion should be more carefully made. First, references are not presented immediately for some of the points raised (e.g., reparative effect in pancreatic cells); second, there is a simultaneous reference to increased insulin concentrations and increased sensitivity of peripheral insulin receptors, effects that are somewhat controversial; why the need for increased insulin together with increase in insulin sensitivity. The way it is written makes it very hard to follow the line of reasoning. Series of studies are mentioning but without establishment of a line of questioning and answering.
RESPONSE: We have completely modified the discussion and related it more clearly and precisely to the results, as you suggested.
6. The effects of GS and BBR in adiponectin’s are discussed mostly taking into consideration studies that measured pro-inflammatory/anti-inflammatory cytokines or other effectors. Why did the authors not measure such factors in plasma from the patients to have a direct inference instead of just referring to literature results?
RESPONSE: Our research measured Res, Vis, Ap, and Ome since, according to the literature, they are the most promising in the context of obesity and type 2 diabetes mellitus. In this version of our manuscript, we modified the discussion to clarify the relevance of the adipokines we studied and minimized the amount of information related to pro- and anti-inflammatory cytokines, as we did not measure them in our patients because this was not one of our objectives.
7. Page 3 line 31 – “In the first study group”; there is no need to start the sentence with this expression; the same for the following expression “while in the second group”. Needs language editing the sentence.
RESPONSE: We have made the changes you suggested.
8. Page 4 lines 150, 151: “Why mentioning “Additionally”; were samples of blood collected differently for analysis performed? Do not see the need for the inclusion of such sentence in the manuscript.
RESPONSE: We agree to eliminate the word "Additionally" since it confuses the meaning of the idea, as you suggested.
9. Throughout the text there are considerable issues with sentences organization. This deters the reader from an intelligible understanding of the manuscript. In particular the discussion section and some narrative parts of the introduction.
RESPONSE: We noticed these mistakes and we appreciated your comment. We have thoroughly reviewed the entire manuscript, especially the introduction and discussion, to correct and improve the writing.

Reviewer 2 Report
Comments and Suggestions for Authors
In this study, the author investigated the effects of Gymnema sylvestre and Berberine on patients with exogenous obesity by measuring the biochemical and body composition parameters and the serum gene expressions of resistin, omentin, visfatin and apelin. The experiment is complete, but I still have the following questions.
1. Why does the author compare the effects of BBR and CS on obesity and hyperglycemia? What are the similarities and differences between these two drugs? Do the effects of these two drugs complement each other? Will the combined treatment of the two achieve better results?
2. Why didn't the experiment set up a placebo control?
3. In obesity classification, what are the classification criteria for Grade I-Grade III?
4. The sentence in lines 64-66 have grammar mistake.
Author Response
In this study, the author investigated the effects of Gymnema sylvestre and Berberine on patients with exogenous obesity by measuring the biochemical and body composition parameters and the serum gene expressions of resistin, omentin, visfatin and apelin. The experiment is complete, but I still have the following questions.
We appreciate your comments, which allowed us to improve our manuscript.
- Why does the author compare the effects of BBR and CS on obesity and hyperglycemia?
RESPONSE: We compared the effects of BBR and GS because both are reportedly effective in managing obesity and lowering glucose; however, the effects of BBR on the gene expression of visfatin, apelin, omentin, and resistin are unknown, and furthermore, they have not been compared. Mexico has a high prevalence of obesity and type 2 diabetes mellitus, and the population tends to prefer natural products and herbs for their management, which increases adherence to treatment. These phytopharmaceuticals are affordable for our population and can complement the management of our patients. This study compares the evidence of their clinical advantages and side effects.
2. What are the similarities and differences between these two drugs?
RESPONSE: Similarly, both GS and BBR have been shown to have antiobesity, antidiabetic and metabolic regulatory effects. Both improve blood glucose control, stimulate insulin secretion and/or improve insulin sensitivity in peripheral tissues, and berberine intervenes in the regulation of lipid metabolism, helping to reduce fat accumulation. However, their mechanisms of action differ, which we show in the following table.
|
|
Antiobesity activity |
Antidiabetic activity |
|
GS |
>Inhibits the absorption of glucose and fats, reducing body weight, and decrease the accumulation of triglycerides in muscles and liver, reducing the accumulation of fatty acids in circulation (1) >Significantly reduces weight gain, mean arterial pressure, leptin, insulin and lipid levels in high-fat diet-induced obese rats (2) >The administration improves glucose metabolism in patients with IGT by significantly decreasing 2-h PG and A1C and increasing insulin sensitivity, in addition to promoting a decrease in BMI and LDL-C (3) |
>Stimulates insulin secretion and regeneration of pancreatic islet cells and improves glucose utilization by activating enzymes related to glucose metabolism since the gymnemic acids present in GS have a structure like glucose, which allows them to bind to receptors in the intestine (1) >Gymnemic acids act by blocking glucose receptors in the intestine, thereby reducing glucose absorption (4) |
|
BBR |
>Modulates AMPK signaling in tissues to improve lipid profile and reduce fat accumulation (5) >It acts on the intestinal microbiota, affecting energy metabolism and preventing obesity induced by high-fat diets (6) >Regulates the production and activity of several adipokines, such as adiponectin, thus improving the metabolic profile and reducing inflammation associated with obesity (7) |
>Increases glucose consumption independently of AMPK activation and improves insulin resistance in adipocytes and suppresses hepatic gluconeogenesis and adipogenesis (5) >Inhibits the hepatic glucagon pathway, crucial for glycemic control in type 2 diabetes, modulates the release of GLP-1 and promotes glycolysis through increasing glucokinase activity (7)
|
- Pothuraju, R.; Sharma, R.K.; Chagalamarri, J.; Jangra, S.; Kumar Kavadi, P. A systematic review of Gymnema sylvestre in obesity and diabetes management. J Sci Food Agric, 2014, 94(5), 834–840. https://doi.org/10.1002/jsfa.6458
- Kumar, V.; Bhandari, U.; Tripathi, C.D.; Khanna, G. Protective Effect of Gymnema sylvestre Ethanol Extract on High Fat Diet-induced Obese Diabetic Wistar Rats. Indian J Pharm Sci, 2014, 76(4), 315–322.
- Gaytán Martínez, L.A.; Sánchez-Ruiz, L.A.; Zuñiga, L.Y.; González-Ortiz, M.; Martínez-Abundis, E. Effect of Gymnema sylvestre Administration on Glycemic Control, Insulin Secretion, and Insulin Sensitivity in Patients with Impaired Glucose Tolerance. J Med Food, 2021, 24(1), 28–32. https://doi.org/10.1089/jmf.2020.0024
- Kang, M.H.; Lee, M.S.; Choi, M.K.; Min, K.S.; Shibamoto, T. Hypoglycemic activity of Gymnema sylvestre extracts on oxidative stress and antioxidant status in diabetic rats. J Agric Food Chem, 2012, 60(10), 2517–2524. https://doi.org/10.1021/jf205086b
- Habtemariam S. The Quest to Enhance the Efficacy of Berberine for Type-2 Diabetes and Associated Diseases: Physicochemical Modification Approaches. Biomedicines, 2020, 8(4), https://doi.org/10.3390/biomedicines8040090
- Guo, H.H.; Shen, H.R.; Wang, L.L.; Luo, Z.G.; Zhang, J.L.; Zhang, H.J.; Gao, T.L.; Han, Y.X.; Jiang, J.D. Berberine is a potential alternative for metformin with good regulatory effect on lipids in treating metabolic diseases. Biomed Pharmacother, 2023, 163, https://doi.org/10.1016/j.biopha.2023.114754
- Xu, X.; Yi, H.; Wu, J.; Kuang, T.; Zhang, J.; Li, Q.; Du, H.; Xu, T.; Jiang, G.; Fan, G. Therapeutic effect of berberine on metabolic diseases: Both pharmacological data and clinical evidence. Biomed Pharmacother, 2021, 133, https://doi.org/10.1016/j.biopha.2020.110984
3. Do the effects of these two drugs complement each other
RESPONSE: Based on our previous findings and our results, we believe that they could have synergistic effects because their mechanisms of action are different and complementary for the therapeutic purpose that we are pursuing in patients with obesity who generally have comorbidities such as insulin resistance, glucose intolerance, and type 2 diabetes mellitus. Therefore, we assume that the combined management of BBR and GS can enhance the control of obesity and even prevent or reduce hyperglycemia in these patients, but further studies are needed.
4. ¿Will the combined treatment of the two achieve better results?
RESPONSE: Due to their mechanisms of action and chemical structure, they could be drugs with a synergistic effect; however, in silico studies are required to predict their effect as a single molecule or as separate molecules in interaction with receptors that activate signaling pathways of interest. We believe that our results lay the foundation for conducting a prospective study in which the possible synergistic effects of combined phytodrugs, as well as their adverse effects on patients with obesity, were tested. To determine the precise underlying mechanism, in future studies, we will find it interesting to perform in silico techniques to understand the interactions of each molecule. We plan to study the HPLC data that describe each molecule present. This will allow us to predict their effect as a single molecule or as separate molecules in interaction with receptors that activate signaling pathways of interest in diabetes and obesity.
5. Why didn't the experiment set up a placebo control?
RESPONSE: We did not have a placebo group because the design of our research was a comparative study between two natural products and not a controlled clinical trial, with the intention to increase originality since the effect of both GS and BBR on obesity is well known; however, the effects of both GS and BBR on obesity have not been studied in our population at the level of modification of adipokines such as resistin, omentin, visfatin, and apelin.
6. In obesity classification, what are the classification criteria for Grade I-Grade III?
RESPONSE: We appreciate your comment. We applied the classification of obesity according to the World Health Organization (https://www.who.int/europe/news-room/fact-sheets/item/a-healthy-lifestyle---who-recommendations). Due to your comment, we realized that it is currently class instead of grade, so we modified it in the manuscript.
7. The sentence in lines 64-66 have grammar mistake.
RESPONSE: We have corrected the grammar mistakes.

Round 2
Reviewer 1 Report
Comments and Suggestions for Authors
Thanks for answering all the raised questions.
Reviewer 2 Report
Comments and Suggestions for Authors
The author provided a detailed answer to my question.